# The Suppressive Effects of Biochar on Above- and Belowground Plant Pathogens and Pests: A Review

**DOI:** 10.3390/plants11223144

**Published:** 2022-11-17

**Authors:** Giuseppina Iacomino, Mohamed Idbella, Stefania Laudonia, Francesco Vinale, Giuliano Bonanomi

**Affiliations:** 1Department of Agricultural Sciences, University of Naples Federico II, 80055 Portici, Italy; 2Center for Studies on Bioinspired Agro-Environmental Technology, BAT Center, University of Naples Federico II, 80055 Portici, Italy; 3Department of Veterinary Medicine and Animal Productions, University of Naples Federico II, 80137 Naples, Italy; 4Task Force on Microbiome Studies, University of Naples Federico II, 80055 Portici, Italy

**Keywords:** plant disease, sustainable control, organic amendment

## Abstract

Soilborne pathogens and pests in agroecosystems are serious problems that limit crop yields. In line with the development of more ecologically sustainable agriculture, the possibility of using biochar to control pests has been increasingly investigated in recent years. This work provides a general overview of disease and pest suppression using biochar. We present an updated view of the literature from 2015 to 2022 based on 61 articles, including 117 experimental case studies. We evaluated how different biochar production feedstocks, pyrolysis temperatures, application rates, and the pathosystems studied affected disease and pest incidence. Fungal pathogens accounted for 55% of the case studies, followed by bacteria (15%), insects and nematodes (8%), oomycetes and viruses (6%), and only 2% parasitic plants. The most commonly studied belowground pathogen species were *Fusarium oxysporum* f. sp. *radicis lycopersici* in fungi, *Ralstonia solanacearum* in bacteria, and *Phytophthora capisci* in oomycetes, while the most commonly studied pest species were *Meloidogyne incognita* in nematodes, *Epitrix fuscula* in insects, and both *Phelipanche aegyptiaca* and *Orobanche crenata* in parasitic plants. Biochar showed suppression efficiencies of 86% for fungi, 100% for oomycetes, 100% for viruses, 96% for bacteria, and 50% for nematodes. Biochar was able to potentially control 20 fungal, 8 bacterial, and 2 viral plant pathogens covered by our review. Most studies used an application rate between 1% and 3%, a pyrolysis temperature between 500 °C and 600 °C, and a feedstock based on sawdust and wood waste. Several mechanisms have been proposed to explain disease suppression by biochar, including induction of systemic resistance, enhancement of rhizosphere competence of the microbial community, and sorption of phytotoxic compounds of plant and/or microbial origin. Overall, it is important to standardize biochar feedstock and the rate of application to improve the beneficial effects on plants in terms of disease control.

## 1. Introduction

Biochar is a heterogeneous material produced by pyrolysis, a thermal process carried out at temperatures between 200 °C and >900 °C and under limited oxygen availability [1].

Biochar differs from charcoal because it can be used as an efficient soil amendment [2]. The elemental composition of biochar varies depending on the biomass feedstock from which it is produced and the characteristics of the pyrolysis process [3]. Biochar is characterized by a high C-to-N ratio, even exceeding 100, and a high content of organic aromatic carbon. Thanks to these properties, biochar is resistant to microbial degradation and its estimated average residence time in the soil varies from centuries to millennia [4]. The use of biochar in agriculture is not new, but dates back to ancient times when the pre-Columbian people of Amazonia developed the so-called “terra preta” or “dark earth” soils through repeated cycles of fire and cultivation, i.e., the slash-and-char system [5]. In this way, nutrient-poor and highly weathered acidic soils were transformed into a fertile substrate that could sustain indigenous populations [6]. Several studies have confirmed the positive interactions of biochar with soil, such as liming effect [7], increasing water retention capacity [8], and the ability to adsorb phytotoxic organic molecules [9]. The changes induced by biochar may well affect nutrient cycling [10] and soil structure [11], thus indirectly affecting plant growth [12], and also soil organic matter cycling [13,14,15]. In addition, biochar has been shown to stimulate the activity of beneficial microbes [12] and suppress soilborne pathogens [16]. The beneficial effects of biochar are often explained by its porosity and sorption capacity [17], redox properties [18,19], and influence on soil structure [20,21].

Soilborne plant pathogens in agroecosystems pose serious problems for agriculture and crop yields. Organic amendments have already been proposed to reduce the incidence of diseases caused by soilborne pathogens [22]. In this context, biochar seems to be a promising tool for controlling various plant pathogens. Indeed, an important application of biochar is its use as an agent for the effective control of plant diseases. Bonanomi et al. [16] reported that biochar effectively suppresses diseases caused by airborne and soilborne plant pathogens such as *Fusarium oxysporum* f. sp. *asparagi*, *F. oxysporum* f. sp. *radicis-lycopersici*, *F. proliferatum*, *Pythium aphanidermatum*, *Phytophthora cactorum*, *P. cinnamomi*, and *Rhizoctonia solani*. Previously, both Elad ad et al. [23] and Harel et al. [24] reported that biochar produced from wood and greenhouse wastes significantly reduced the incidence of powdery mildew caused by *Leveillula taurica* on *Lycopersicon esculentum* and *Podosphaera aphanis* on *Fragaria x ananassa*, respectively. Five main mechanisms have been proposed to explain disease suppression by biochar: (i) induction of systemic resistance in host plants; (ii) enhanced abundance and activity of beneficial microbes, including mycorrhizal fungi; (iii) alteration of soil quality in terms of nutrient availability and abiotic conditions, such as liming effect; (iv) direct fungitoxic effect of biochar; (v) sorption of allelopathic phytotoxic compounds that can directly damage plant roots and thus promote pathogen infestation. With the aim of developing more ecologically sustainable agriculture, the possibility of using biochar to defend against pathogens has increased in prominence in recent years in light of previous studies. In this context, we provide a general overview of disease suppression via organic amendment with biochar and try to identify the mechanisms underlying this suppression. To this end, this review examines a total of 61 scientific articles including 117 experimental case studies conducted over the period of 2015 to 2022. This review represents an update of previous reviews [16,24] of the state of the art since 2015 regarding the potential of biochar as a soil disease control agent. Specifically, we focus on three main aspects that affect disease suppression and pest control: (i) type and quality of production feedstock, (ii) temperatures, and (iii) application rates. We also provide an analysis of the mechanisms behind the suppressive effects of biochar. Overall, this review allows a fair and complete assessment of the suppressive effects that could be expected when using biochars produced with different feedstocks and pyrolysis conditions, which possible side effects must be considered, and to what extent biochar application may promise benefits in the management of agronomic disease.

## 2. Results

### 2.1. Research Efforts over the Last Seven Years: Pathosystem, Feedstock Type, and Pyrolysis Conditions

In 117 case studies reviewed, we observed that most of them (41%) were based on *Solanum lycopersicum*, followed by *Jatropha curcas* (6%) and *Panax notoginseng* (6%), *Capsicum annum* (4%), *Solanum tuberosum* (4%) and *Lactuca sativa* (4%), *Cucumis sativus* (3%), *Morus alba* (3%), *Asparagus officinalis* (3%), and *Glycine max* (3%). Finally, the remaining 23% of cases were based on other species, e.g., *Lolium perenne*, *Malus commuins*, *Cucumis melo*, *Manihot esculenta*, *Phaseolus vulgaris*, *Daucus carot*, *Allium cepa*, *Oryza sativa*, *Zea mays*, *Radix pseudostellariae*, *Fragaria* sp., *Lepidium sativu*, *Solanum melongena*, *Nicotiana tabacum*, *Brassica rapa*, and *Lupinus angustifolius* (Figure 1A, Appendix A). Overall, we found that 55% of the case studies concerned pathogenic fungi, 15% bacteria, 8% insects, 8% nematodes, 6% oomycetes, 6% viruses, and 2% parasitic plants (Figure 1B, Appendix A). The most studied species among the fungi was *Fusarium oxysporum* f. sp. *radicis lycopersici*, followed by *Rhizoctonia solani*, *Botrytis cinerea*, *Fusarium oxysporum*, *Alternaria solani*, *Fusarium solani*, *Sclerotinia sclerotiorum*, and *Fusarium* spp. (Figure 1C). *Ralstonia solanacearum* was the most investigated species among the bacteria; for nematodes, insects, oomycetes, and viruses, the most studied species were *Meloidogyne incognita*, *Epitrix fuscula*, *Phytophtora capisci*, and Tomato yellow leaf curl virus, respectively. Finally, for parasitic plants, both *Phelipanche aegyptiaca* and *Orobanche crenata* were investigated (Figure 1C, Appendix A). 

Concerning the pyrolysis process, the most used pyrolysis temperature range among our examined study cases was between 500 °C and 600 °C, followed by 300–400 °C and 600–700 °C (Figure 2A). Finally, regarding feedstock types used in the collected research articles, the most used were sawdust and wood residues, followed by grass residues (straw and stalk) and forb crop residues (Figure 2B).

### 2.2. Disease Suppression by Biochar

Overall, biochar treatments showed an effective disease-suppressive effect of 86% for fungi, 100% for oomycetes, 100% for viruses, 96% for bacteria, and 50% for nematodes, compared to untreated controls. In a few cases, an increase in disease incidence was observed after biochar application. The jitter box plot in Figure 3 shows that the rate of disease suppression against bacterial, fungal, oomycete, and viral pathogens was rather variable among the case studies. For bacteria, in fact, very high variability was observed between the case studies in which the biochar had a negative effect on disease suppression, i.e., a conducive effect, and the case studies in which there was a positive response, i.e., a suppressive effect. Many studies have demonstrated the efficacy of biochar against the bacteria present in soil, especially against *Ralstonia solanacearum* [25]. Fungi also varied widely in their responses in terms of disease suppression. In studies of airborne pathogenic fungi, application of various wood-derived biochars was able to control disease caused by *Alternaria solani* [26] and *Botrytis cinerea* on *Solanum lycopersicum* [27] and *Fragaria x ananassa* [28]. Biochars derived from green waste, straw, and husks effectively controlled *A. solani* [26], *B. cinerea* [28], *Leveillula taurica* [29], *Magnaporthe oryzae* [30], *Phyllactinia corylea*, and *Pseudocercospora mori* [31].

The ability of biochar to suppress soilborne pathogens was reported for *Fusarium oxysporum* f. sp. *radicis lycopersici* [32], *Fusarium* spp. [33], *Fusarium oxysporum* [34], *Fusarium oxysporum* f. sp. *asparagi*, *Fusarium proliferatum* [35], *Fusarium verticilloides* [36], *Fusarium solani* [37], *Verticillium dahliae* [38], *Sclerotinia sclerotiorum* [25], *Rhizoctonia solani*, *Macrophomina phaseolina*, *Sclerotium cepivorum*, *Sclerotium rolfsii* [34], and *Fusarium torulosum* [25]. Finally, considerable variability in the disease-suppressive effect of biochar was also observed in oomycetes and viruses. Overall, the present findings indicate that biochar can potentially control 20 fungal, 8 bacterial, and 2 viral plant pathogens.

### 2.3. Application Rate

Based on correlation analysis, our results show that there is a strong positive correlation between disease-suppression rate and biochar application rate for the most studied microbial domains, i.e., fungi (r = 0.45, *p* < 0.001), bacteria (r = 0.46, *p* = 0.032), and oomycetes (r = 0.96, *p* < 0.001) (Figure 4). Moreover, our results show that the range of biochar application in the 117 case studies examined was from 1% to 10% for fungi, from 2% to 5% for bacteria, from 1% to 3% for oomycetes, from 1% to 5% for nematodes, from 1% to 3% for viruses, and from 1% to 20% for insects. In most case studies, the application range was between 1% and 3%. Regardless of application rate, the results showed varying responses to disease suppression, ranging from −50% to 100%, suggesting that pathogen response to biochar depends on a combination of several factors and not just on application rate, as we found a high variability of disease suppression among studies that used similar application rates.

### 2.4. Feedstock Type and Pyrolysis Temperature

Our analysis of the link between disease-suppression rate and pyrolysis temperature showed a unimodal relationship (Figure 5A). The most commonly used temperatures ranged from 350 °C to 600 °C. The application of biochar produced at a temperature between 350 °C and 600 °C showed the most positive effect on disease suppression, although a high variability was found among case studies. When biochar was produced at lower or higher temperatures, some negative effects were observed regarding disease suppression. On the other hand, our results showed that the most common raw material used in the case studies, wood residues, affected disease control very differently, with both very low and high suppression efficacies observed (Figure 5B). Straw and husks also strongly affected disease suppression, with no negative effect observed. In the case of raw hull materials, cases were observed in which disease control was only positive. Finally, two cases with no effect and one case with a positive response in terms of disease control were observed for raw bark materials.

### 2.5. Disease-Suppression Putative Causal Mechanisms

In the 117 cases studied, four different mechanisms were proposed to explain how biochar controls plant diseases: (i) adsorption/deactivation of virulence factors; (ii) induced and acquired systemic resistance (ISR, SAR); (iii) alteration of the soil microbial community; and (iv) increased soil Si content.

Adsorption/deactivation of virulence factors was noted as a suspected mechanism of the disease-suppressive effect of biochar in 12 case studies included within our review (Table 1). Pathogens present in the soil produce several extracellular cell-wall-degrading enzymes (CWDEs) and a number of toxic metabolites in plants, including mycotoxins and antibiotics. Pectinolytic and cellulolytic enzymes are the main enzymes secreted by soil pathogens [39]. Due to its ability to adsorb small and large organic compounds, including enzymes [40], the application of biochar to soil helps to reduce the availability of such compounds. This adsorption capacity is explained by the low specific surface area and very high mineral content of biochar. For example, Jaiswal et al. [32] suggested that biochar adsorption of CWDE products and toxic metabolites produced by soilborne pathogens may help to protect plant roots from pathogen attack, a hypothesis originally proposed by Graber et al. [41].

On the other hand, induced and acquired systemic resistance (ISR and/or SAR) was suggested as a mechanism by which biochar can reduce disease incidence in nine case studies. Jaiswal et al. [42] showed in their study that biochar prepared from greenhouse plant waste induced systemic resistance *to F. oxysporum* f. sp. *radicis lycopersici* while improving the growth and physiological parameters of tomato plants by up to 63%. Two main forms of induced resistance (IR) have been described in plants, induced systemic resistance (ISR) and systemic acquired resistance (SAR). SAR and ISR are mainly differentiated on the basis of the elicitor and the regulatory pathways involved. ISR depends on ethylene (ET) and jasmonic acid (JA) phytohormones, while SAR is mediated by the phytohormone salicylic acid (SA) [43]. In addition, SAR is characterized by the activation of a large set of pathogenesis-related (PR) genes, which are involved in defense responses, while ISR functions without PR gene activation. Jaiswal et al. [42] showed via transcriptomic analysis (RNA-seq) of tomato that biochar upregulates genes associated with plant defense and growth, such as jasmonic acid, brassinosteroids, cytokinins, auxin, and phenylpropanoids. In contrast, genes associated with the biosynthesis and signal transduction of salicylic acid metabolism were largely downregulated. Regarding the induction of acquired and induced resistance, 12 case studies investigating these mechanisms are reported within our review (Table 1). Nevertheless, we found a large gap in the literature regarding the identification of the specific genes activated by biochar and involved in the induction of systemic resistance in plants.

Concerning the role of the soil microbiome, 73 case studies reported the alteration of soil microbial community as a possible mechanism of disease suppression. The population structure of soil microbes is critical for soil function and ecosystem services, affecting soil structure and stability, nutrient cycling, aeration, water use efficiency, disease resistance, and organic carbon storage capacity [44]. In fact, beneficial microbiota can compete with pathogens for space and nutrients or produce microbial agents to improve plant health [45]. Biochar application greatly affects soil physiochemical properties and thus microbial communities and processes. Lu et al. [46] demonstrated that biochar application significantly reduced the severity of bacterial wilt caused by *R. solanacearum*, possibly due to an increase in the population density of soil bacteria and Actinobacteria, would have led to improved resistance to the pathogen. Specifically, adding biochar to pathogen-infected soil significantly increased the density of soil bacteria and Actinobacteria, but decreased fungal density and the ratios of fungi/bacteria and fungi/Actinobacteria in the soil. Biochar treatment also increased the soil neutral phosphatase activity and urease activity. Moreover, Rasool et al. [26] used biochar in combination with a compost mixture of plant-growth-promoting rhizobacteria (PGPR), which stimulated rhizobacterial activity, resulting in plant growth activation and disease inhibition. In addition, Bonanomi et al. [47] reported that biochar in combination with alfalfa or manure was among the most effective soil treatments for controlling Tomato spotted wilt virus-caused disease on tomato plants. The authors speculated that the suppressive effect of biochar was due to the support of known beneficial microbes in the soil, such as Actinobacteria. De Tender et al. [28] proposed that biochar protects the rhizosphere community from pathogen spread in three ways: (1) by increasing the richness of the bacterial community, (2) by increasing the relative abundance of genera, including species that act as biocontrol agents or are involved in nitrogen cycling, and (3) by shifting the relative abundance of bacterial genera in the rhizosphere to those received after plant infection. The latter mechanism is the most common hypothesis. Although we found that 73 case studies showed that biochar acts through microbiome changes to suppress soil diseases, no study defined which specific microorganisms were involved. Therefore, we suggest that future research should focus on filling this gap.

Increasing soil silicon (Si) content is another putative mechanism suggested in our review, particularly in two studies, to suppress soilborne diseases. Si is an important nutritional element for plant growth, and Si in plant tissues protects against biotic and abiotic stresses such as toxic metal stress, resistance to microbial pathogens in leaves, and drought tolerance [48,49]. Rice is considered a Si-rich crop that can take up silicon from the soil as soluble silica, and rice shoots contain up to 10% Si in dry matter [50]. Chen et al. [51] demonstrated that adding biochar to soil prolonged larval development, increased larval mortality, decreased larval body weight, and reduced adult *Cnaphalocrocis medinalis* longevity. This developmental performance suggests that application of biochar to rice may negatively affect *C. medinalis*. One possible interpretation is that biochar amendment in the soil increases the resistance of silicon-associated plants to herbivores. The application of biochar to soil increases the availability of Si in the soil and the uptake of Si by rice plants [52]. Higher Si content in a host plant can increase not only physical resistance but also chemical defenses triggered by herbivores [53]. In *C. medinalis*, soil Si enrichment can reduce feeding damage to rice plants by leaf-bending larvae [54].

## 3. Discussion

Apart from its direct positive effects on plant growth and promotion [55], biochar appears to be a new and promising tool for controlling several plant diseases and pests, with most case studies reporting positive suppressive effects. Studies investigating the effects of different types of biochar on plants are summarized below and subdivided by pathogen type.

### 3.1. Fungi and Oomycetes

Fungi and oomycetes can be divided into two broad groups: obligate parasites, which depend entirely on living host plant tissue for nutrition and reproduction, and facultative parasites, which cause significant damage to plants but can also live as saprophytes on plant debris and organic material [56]. Pathogens that attack aboveground plant organs are referred to as foliar pathogens, while those that attack the root system and reside primarily in the soil are referred to as soilborne pathogens [57]. The potential benefits of biochar in suppressing diseases caused by pathogenic soilborne fungi have been demonstrated in several studies (Table 2) For example, Akhter et al. [58] evaluated the response of *Fusarium oxysporum* f. sp. *lycopersici* chlamydospores on tomato plants grown in soil enriched with biochar and compost and found that the amended soil had great potential to suppress chlamydospore infectivity and reduce pathogen-related physiological stress in tomato plants. Moreover, Akanmu et al. [36] demonstrated the efficacy of biochar in controlling *Fusarium* ear rot in maize. Similar results were reported by Wu et al. [59], who found that soil treatment with biochar resulted in a reduction in the abundance of *Fusarium oxysporum* and a reduction in the virulence of the fungus on *Radix pseudostellariae* plants. The suppressive effect of biochar was also tested against airborne plant pathogens. For example, Rasool et al. [26] studied the effect of green waste biochar (GWB) and wood biochar (WB) together with compost and plant-growth-promoting rhizobacteria (PGPR; *Bacillus subtilis*) on the physiology of tomato (*Solanum lycopersicum*) and the development of *Alternaria solani*, and showed for the first time that disease suppression was strongest (up to 80%) in the presence of *B. subtilis* in the GWB-containing substrate. In addition, De Tender et al. [28] showed how biochar treatments can improve the disease resistance of strawberry plants to the airborne fungal pathogen *Botrytis cinerea* by recruiting microbes from the rhizosphere. On the other hand, several studies have also investigated the ability of biochar to suppress oomycetes. Wang et al. [60] investigated the suppression of *Phytophthora pepper* blight in a pot experiment as a function of time after biochar application. Biochar treatment effectively inhibited pathogen growth, reduced disease by up to 91%, and significantly increased the incidence of potential biocontrol fungi. However, a few case studies have shown a negative effect of biochar on disease control. For example, Copley et al. [61] showed that maple bark biochar increased soybean susceptibility to diseases caused by *Rhizoctonia solani*. At lower concentrations (1% and 3%), biochar was ineffective against the disease, but at a 5% application rate, biochar treatment showed a significant increase in disease severity caused by *R. solani.* The authors provide compelling evidence that biochar is associated with the downregulation of a number of genes related to primary and secondary plant metabolism, such as genes involved in amino acid metabolism, cell wall plasticity, and the tricarboxylic acid (TCA) cycle, which likely facilitated entry points, resulting in higher susceptibility to *R. solani*.

### 3.2. Bacteria

Regarding pathogenic bacteria in soil, a suppressive capacity of biochar was reported for *Ralstonia solanacearum* [25], *Kosakonia sacchari* [59], *Agrobacterium tumefaciens* [62], and *Streptomyces scabies* [63]. Regarding airborne pathogenic bacteria, Bonanomi et al. [62] reported a high efficacy of biochar against *Pseudomonas syringae* and *Pseudomonas viridiflava* on *Solanum lycopersicum* plants. No case studies of negative effects of biochar on the suppression of diseases caused by bacteria were reported (Table 3). In contrast, Tian et al. [25] investigated the efficacy of wheat straw biochar for control of bacterial wilt of tomato caused by *R. solanacearum.* Their results showed that biochar reduced disease incidence by 61% to 78% compared to the control without biochar while improving plant growth, likely due to increased microbial activity and changes in organic matter and amino acid composition in the rhizosphere. Furthermore, Gu et al. [64] investigated the efficacy of applying 3% wood biochar to suppress bacterial wilt of tomato caused by *R. solanacearum.* Specifically, the application of fine biochar significantly reduced the incidence of bacterial wilt by 20% and reduced pathogen mobility and rhizosphere colonization.

### 3.3. Viruses

Little work has been done on the effects of biochar soil amendment on plant viruses [47,65,66] (Table 4). However, the few studies available show great potential to protect plants from phytopathogenic viruses using biochar.

Specifically, Zeshan et al. [66] tested the efficacy of biochar from maize at 1%, 2%, and 3% concentrations on tomato (*Solanum lycopersicum*) plants infected with leaf curl virus. After biochar soil treatment, disease severity was found to be 22%, which was significantly lower than that of the control (40%). Kawanna et al. [65] reported a reduction in tomato mosaic virus incidence after the application of 1% and 1.5% biochar obtained from rice husk material. The rates of infection and disease severity were reduced by 50% and 37% following treatments with 1.5% and 1% biochar in the soil, respectively. Finally, Bonanomi et al. [47] studied the suppressive effect of biochar against Tomato spotted wilt virus in tomato plants. Plants grown in soils treated with biochar had a lower incidence of the disease (<40%) than those grown in soils treated with mineral fertilizer and fumigation (>80%). In their study, the authors noted a significant change in soil microbial community and structure after biochar application, and speculated that induction of resistance might be the cause of disease suppression.

### 3.4. Nematodes and Insects

Arshad et al. [67] tested biochar derived from rice husks in combination with biological control agents (BCAs) such as *B. subtilis* and *Trichoderma harzianum* against *Meloidogyne incognita* in tomato (*S. lycopersicum*). The results indicate that applying 3% biochar with BCAs effectively controlled root-knot nematode, improved overall plant biomass, and activated genes related to tomato plant defense. Similarly, Oche Eche and Okafor [68] showed that biochar from gum arabic, bush mango, and locust bean is a promising control agent for *M. incognita*. In addition, Marra et al. [69] demonstrated that biochar from olive mill waste produced complete inhibition of the root-knot nematode *M. incognita* due to the presence of several compounds in the biochar, mainly fatty acids and phenols, which are known to be among the phytochemical compounds that exhibit nematicidal effects.

Regarding insects, Chen et al. [51] investigated the effect of biochar on the development and reproductive performance of *Cnaphalocrocis medinalis* on rice and examined the population size of *C. medinalis*, showing that biochar can affect its development and has negative effects on its population. Furthermore, Edenborn et al. [70] studied the effects of modified hardwood biochar with different types of compost tea and microbial enrichment from vermicompost on eggplant (*S. melongena* var. Rosa Bianca) growth and flea beetle (*Epitrix fuscula*) damage. The authors found that adding biochar did not improve resistance to insect damage.

### 3.5. Parasitic Plants

Parasitic plants are a taxonomically diverse group of angiosperms that are partially or completely dependent on host plants for carbon, nutrients, and water, which they obtain by attaching to the roots or shoots of the host. Parasitism often results in severely impaired host plant performance, leading to changes in the competitive relationships between host and nonhost plants and a cascade of effects on community structure and diversity, vegetation cycling, and zonation [71]. Our research revealed two case studies of use of biochar against parasitic plants. Eizenberg et al. [72] conducted experiments in pots of tomato (*S. lycopersicum*) infested with *Phelipanche aegyptiaca* (Egyptian broomrape) using biochar prepared from greenhouse pepper plant waste. The addition of biochar resulted in reduced infestation of the tomato plants, mainly by reducing germination of *P. aegyptiaca* seeds due to adsorption to the biochar of stimulatory molecules, i.e., strigolactones. On the other hand, Saudy et al. [73] conducted the first field experiment investigating the use of biochar for control of broomrape weed (*Orobanche crenata*) in two faba bean cultivars. Biochar prepared from dry plant waste of *Casuarina equisetifolia* was associated with significant reduction of broomrape infestation. The authors suggested that the addition of biochar might represent a barrier handicapping the accession of faba bean root stimulants to broomrape seeds, preventing their germination. Moreover, application of biochar could change the rhizospheric environment to one unsuitable for broomrape seed germination, or could even cause damage to germinated seeds. These studies prove that biochar can also reduce infestations of parasitic weeds in important crops, suggesting new treatment strategies for this type of pest and highlighting the economic feasibility of using biochar. 

## 4. Materials and Methods

### Data Collection and Analysis

To collect data for this review, we searched for scientific articles published between 2015 and 2022 that specifically addressed the suppressive effects of biochar on plant pathogens and pests. The search was conducted in all international journals in the Web of Science and Google Scholar databases using keywords such as “biochar”, “disease suppression”, “plant disease”, “organic amendment”, “pest”, and “soilborne and airborne pathogens”. This search resulted in 61 articles that met our conditions for review. The next step was to collect the data from each article into a spreadsheet. There were 117 case studies in total, taking into account the fact that some articles examined more than one target plant species or pathogen. Specifically, we collected information regarding the following factors from each case study: host plant, pathogen species, pest species, feedstock type, pyrolysis temperature, biochar application rate, disease incidence, and the mechanism of action of the biochar against the pathogen. Our review included studies of pathogenic fungi, bacteria, nematodes, insects, oomycetes, viruses, and parasitic plants. The feedstock type refers to the material used to produce the biochar under pyrolysis conditions, and the application rate accounted for the quantity of biochar applied. The incidences of diseases and infestations were recorded as the percentage by which the biochar reduced them compared to the control. After the corresponding tables were prepared, a correlation plot between disease incidence and biochar application rate for the most important pathogen types was generated using the cor function in the stats package, and the *p*-value was calculated using the cor.test function in the stats package. The level of significance for all statistical tests was set at *p* ≤ 0.05.

## 5. Conclusions

The use of biochar has been shown to have beneficial effects on disease control, although these effects vary in their efficacy and mode of action. The properties of biochar and the associated effects on pathogen control depend mainly on the feedstock biomass, pyrolysis conditions, and application rate. Our review highlighted that the best disease control results were obtained with biochar produced from sawdust and wood residues at pyrolysis temperatures between 350 °C and 600 °C. However, it appears to be a drawback that the efficacy of biochar depends on these conditions, as the production and application of biochar would have to be specific in each case [74]. Furthermore, most of the studies described in our review showed that biochars of different origins can be effective against a range of pathogens and pests. The efficacy of biochar in disease control implies different mechanisms, such as the adsorption of virulence factors, the triggering of systemic resistance in plants, the alteration of the soil microbial community, and/or the increase in soil silicon content.

Several of the experiments conducted have already expanded our understanding of the mechanisms underlying the effects of different types of biochar on disease suppression. However, our understanding of the effects of biochar on soilborne disease progression is still in its infancy. We hypothesize that the positive impact of biochar addition on soil microbial structure and diversity is one of the keys to the effect of biochar. Nevertheless, further research is needed to determine the microbial species that are affected by biochar and that interact with diseases to reduce them. In addition, most of the disease organisms tested were fungi or related organisms. 

Further studies on feedstocks and optimized administration methods are needed to understand biochar’s effects and mechanisms of action in the control of arthropod infestations and diseases caused by bacteria, viruses and nematodes. Moreover, recent studies have indicated repellent effects of biochar byproducts such as smoked water against arthropods pests [75]. Moreover, special attention needs to be paid to the longevity of biochar’s effects in the soil to determine whether these effects will persist in the field across seasons. Finally, one of the biggest challenges is to optimize the use of biochar by finding an effective biochar type and application rate that will positively affect a wide range of pathosystems.

## Figures and Tables

**Figure 1 plants-11-03144-f001:**
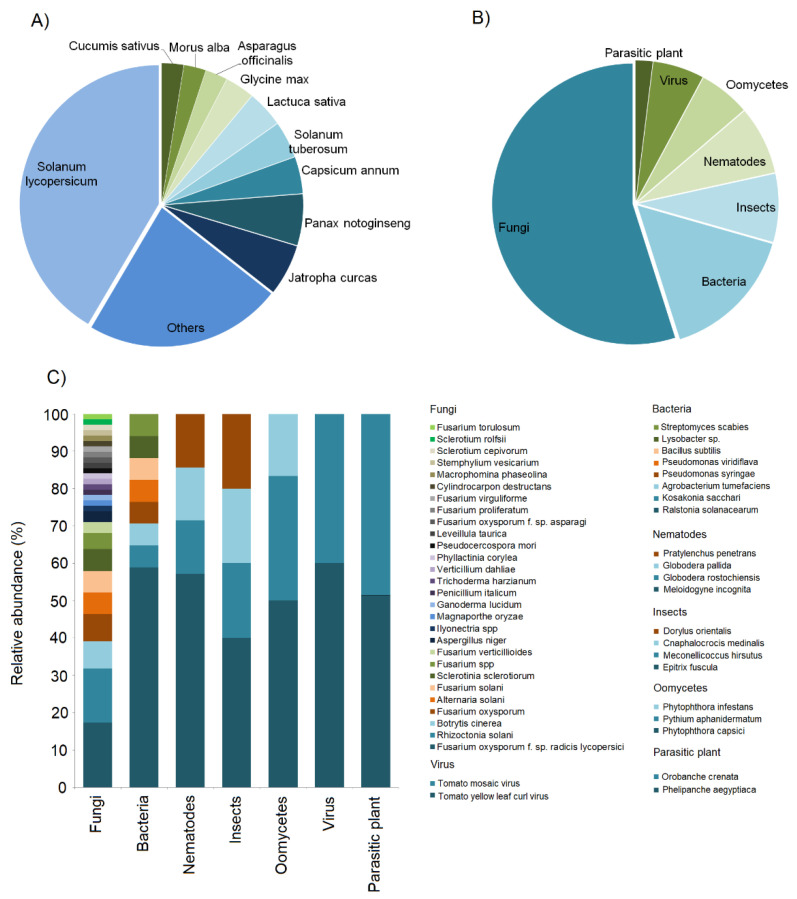
Pie charts representing the proportion of plants (**A**) and pathogen and pest types (**B**) in the systems investigated regarding biochar for pest control and disease suppression in the last seven years. (**C**) Stacked bar charts represent the relative abundances of studied pathogens and pests at the species level.

**Figure 2 plants-11-03144-f002:**
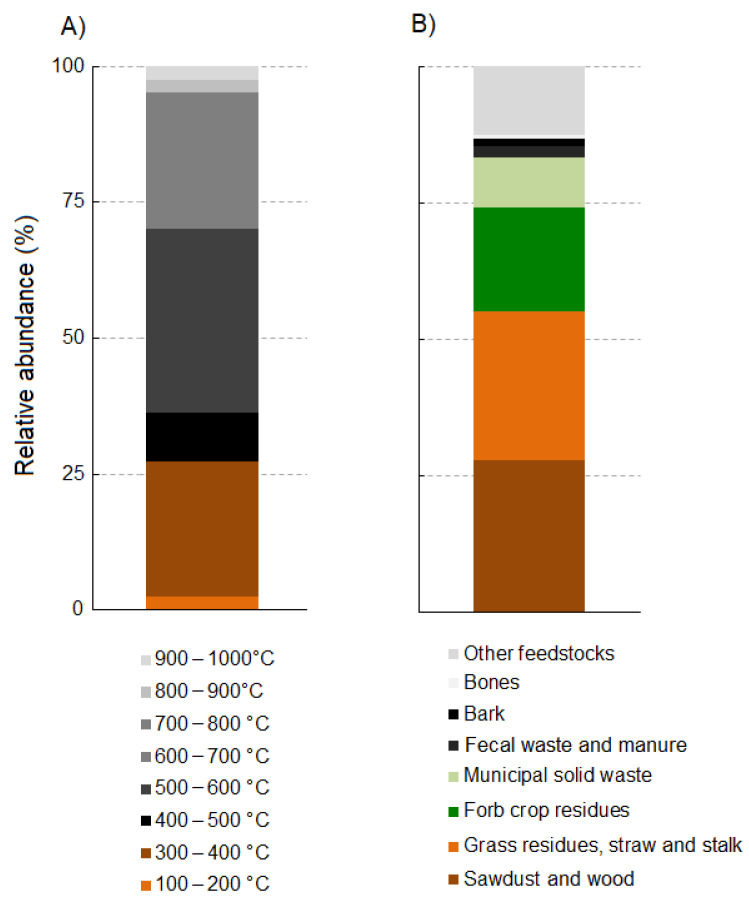
Stacked bar plots showing the relative frequencies of pyrolysis temperature ranges (**A**) and feedstock types (**B**) used in the collected research articles from the last seven years concerning the effects of biochar on plant disease suppression and pest control.

**Figure 3 plants-11-03144-f003:**
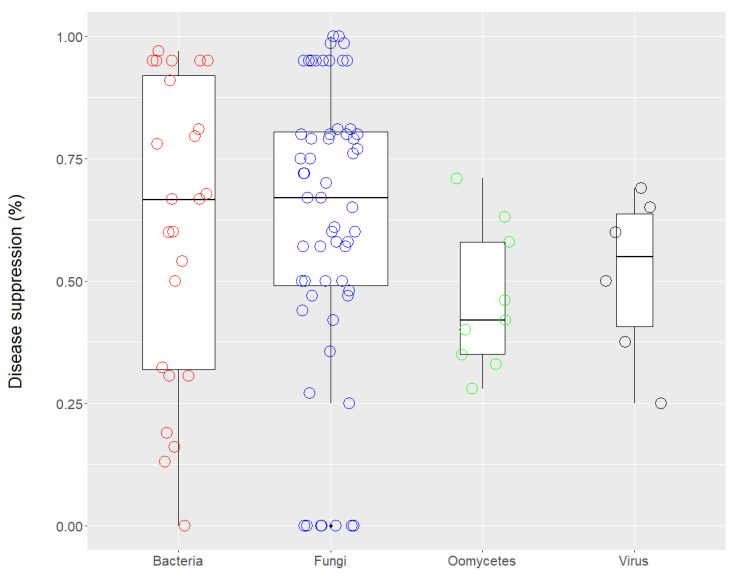
Jitter box plots of the disease-suppression rate against bacterial, fungal, oomycete, and viral pathogens. The lower and upper bounds of the boxplots show the first and third quartiles (the 25th and 75th percentiles); the middle line shows the median; whiskers above and below the boxplot indicate the interquartile range.

**Figure 4 plants-11-03144-f004:**
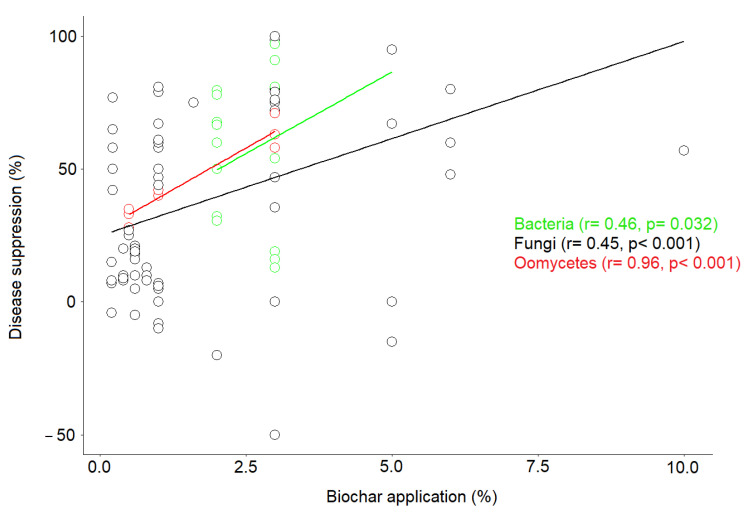
Relationship between disease-suppression rate and biochar application rate recorded for the three most common pathogen types i.e., fungi, bacteria, and oomycetes. The solid-colored lines show a fit by linear regression for each pathogen type.

**Figure 5 plants-11-03144-f005:**
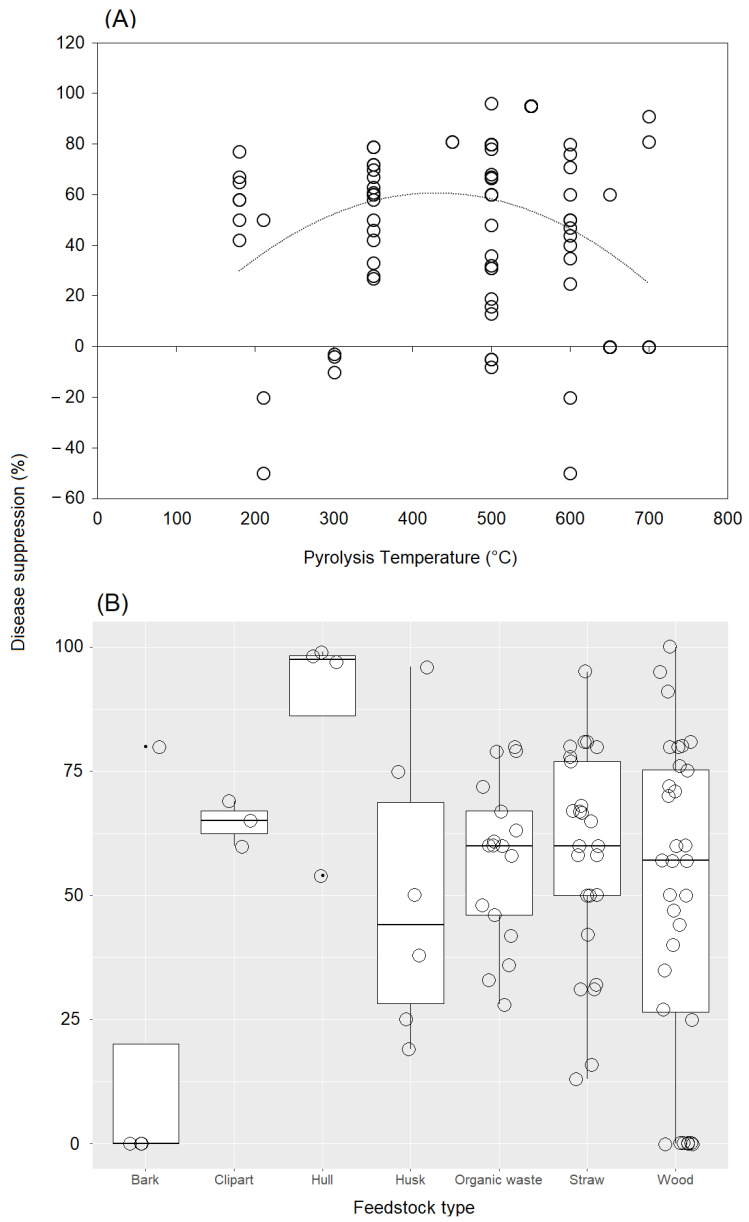
(**A**) Scatter diagram of the relationship between disease-suppression rate and pyrolysis temperature. The dashed black line shows a fit by logarithmic regression. (**B**) Jitter box plots of the disease-suppression rate for each feedstock type. The lower and upper bounds of the boxplots show the first and third quartiles (the 25th and 75th percentiles); the middle line shows the median; whiskers above and below the boxplot indicate the interquartile range.

**Table 1 plants-11-03144-t001:** Count of case studies reporting different putative causal mechanisms by which biochar induces disease suppression or pest control.

Pathogen and Pest Type	Adsorption/Deactivation of Virulence Factors	ISR, SAR	Alteration of Soil Microbial Community	Increased Si Content
Fungi	7	3	50	1
Bacteria	2	2	13	0
Insects	0	0	1	1
Nematodes	0	4	1	0
Oomycetes	0	0	6	0
Viruses	0	3	2	0
Parasitic plants	3	0	0	0

**Table 2 plants-11-03144-t002:** List of experimental studies examples that applied biochar as a soil amendment to control plant diseases caused by airborne (A) and soilborne fungal (SB) pathogens. Pathogen, host plant, biochar feedstock type, response level, and reference are reported for each study.

Pathogen	Host Plant	Biochar Type and Application Rate	Response	References
*Fusarium oxysporum* f. sp. *radicis lycopersici* (SB)	*Solanum lycopersicum*	Wood (1–3%)	Medium–high	[32]
*Alternaria solani* (A)	*Solanum lycopersicum*	Wood—green wastes (3–6%)	high	[26]
*Fusarium oxysporum* f. sp. *radicis lycopersici* (SB)	*Solanum lycopersicum*	Wood—wastes (1–3%)	Medium–high	[32]
*Fusarium oxysporum* f. sp. *radicis lycopersici* (SB)	*Solanum lycopersicum*	Wood—wastes (3%)	high	[58]
*Botrytis cinerea* (A)	*Solanum lycopersicum*	Wood (1%)	high	[27]
*Fusarium oxysporum* f. sp. *radicis lycopersici* (SB)	*Solanum lycopersicum*	Green house wastes (1–3%)	high	[42]
*Alternaria solani* (A)	*Solanum lycopersicum*	Wood—green wastes (3–6%)	Medium–low	[26]
*Fusarium oxysporum* f. sp. *lycopersici* (SB)	*Solanum lycopersicum*	Wood (0.5–3%)	Medium–high	[32]
*Fusarium verticillioides* (SB)	*Zea mays*	Wastes (1–3%)	Medium	[36]
*Fusarium oxysporum* (SB)	*Radix pseudostellariae*	Hull (3%)	High	[59]
*Botrytis cinerea* (A)	*Fragaria x ananassa*	Organic matter (1–3%)	High	[28]
*Fusarium* spp. (SB)	*Panax notoginseng*	Wood (8 g L^−1^)	Medium	[33]
*Magnaporthe oryzae* (A)	*Lolium perenne* L.	Straw (0.22–1%)	Medium–high	[37]
*Fusarium solani* (SB)	*Malus*	Husk (5–80 g kg^−1^)	Medium	[37]
*F.oxysporum* f. sp. *radicis-lycopersici* (SB), *Botrytis cinerea* (A), *Fusarium oxysporum* (SB), *Ganodema lucidum* (SB), *Penicillium italicum* (A), *Rhizoctonia solani* (SB), *Sclerotinia sclerotiorum* (SB)	*Solanum lycopersicum*	Medicago-Mays—organic wastes (5%)	High	[62]
*Verticillium dahliae* (SB)	*Solanum melongena*	Husk (10 t/ha biochar)	High	[38]
*Phyllactinia corylea* (A), *Pseudocercospora mori* (A)	*Morus alba*	Husk	Medium	[31]
*Leveillula taurica* (A)	*Capiscum annum* L.	Green house wastes—wood	High	[29]
*Fusarium oxysporum* (SB), *Fusarium oxysporum* f.sp. *asparagi* (SB), *Fusarium proliferatum* (SB)	*Asparagus officinalis* L.	Wood (10%)	Medium–high	[35]
*Rhizoctonia solani* (SB)	*Glycine max*	Wood (1–5%)	Negative effect	[61]
*Fusarium oxysporum* f. sp. *radicis-lycopersici* (SB), *Fusarium oxysporum* (SB), *Rhizoctonia solani* (SB), *Sclerotinia sclerotiorum* (SB), *Macrophomina phaseolina* (SB), *Sclerotium cepivorum* (SB), *Sclerotium rolfsii* (SB)	*Jatropha curcas* L.	Sewage sludge (0.2–1%)	Medium–high	[34]
*Fusarium torulosum* (SB), *Fusarium solani* (SB)	*Panax ginseng*	Straw (1%)	High	[25]

**Table 3 plants-11-03144-t003:** List of experimental studies examples that applied biochar as a soil amendment to control plant diseases caused by airborne (A) and soilborne (SB) bacterial plant pathogens. Pathogen, host plant, biochar feedstock type, response level, and reference are reported for each study.

Pathogen	Host Plant	Biochar Type and Application Rate	Response	References
*Ralstonia solanacearum* (SB)	*Solanum lycopersicum*	Straw (2%)	High	[25]
*Ralstonia solanacearum* (SB)	*Solanum lycopersicum*	Straw (2%)	High	[46]
*Ralstonia solanacearum* (SB)	*Solanum lycopersicum*	Straw (2%)	Medium–high	[64]
*Kosakonia sacchari* (SB)	*Radix pseudostellariae*	Hull (3%)	High	[59]
*Agrobacterium tumefaciens* (SB)	*Lactuca sativa*	Medicago, wood, organic wastes, maize (5%)	High	[62]
*Pseudomonas syringae* (A)	*Solanum lycopersicum*	Medicago, wood, organic wastes, maize (5%)	High	[62]
*Pseudomonas viridiflava* (A)	*Solanum lycopersicum*	Medicago, wood, organic wastes, maize (5%)	High	[62]
*Lysobacter* sp. (SB)	*Solanum lycopersicum*	Medicago, wood, organic wastes, maize (5%)	High	[62]
*Ralstonia solanacearum* (SB)	*Nicotiana Tabacum*	Hull (7.5–45 ton/ ha)	High	[51]
*Ralstonia solanacearum* (SB)	*Solanum lycopersicum*	Wood (3%)	High	[64]
*Streptomyces scabies* (SB)	*Solanum tuberosum*	Different agricultural wastes (0.5 ton/ha)	Medium	[63]

**Table 4 plants-11-03144-t004:** List of experimental studies that applied biochar as a soil amendment to control plant diseases caused by viral plant pathogens. Pathogen, host plant, biochar feedstock type, response level, and reference are reported for each study.

Pathogen	Host Plant	Biochar Type and Application Rate	Response	References
Curl virus (SB)	*Solanum lycopersicum*	Maize (1–3%)	High	[66]
Tomato Mosaic virus (SB)	*Solanum lycopersicum*	Husk (0.5–1.5%)	Medium	[65]
Tomato spotted wilt virus (TSWV) (SB)	*Solanum lycopersicum*	Wood	High	[47]

## Data Availability

Data can be provided upon private request.

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
