# Peer review of "The Suppressive Effects of Biochar on Above- and Belowground Plant Pathogens and Pests: A Review"

_plants, 2022, doi:10.3390/plants11223144_

Round 1

Reviewer 1 Report

-        Abstract:

-        Line 20-22- It would be more accurate to refer the pathogens and pests as above ground and below ground pathogens and pests.

-        Line 23-24: Please discuss results with some more details

-        Introduction:

-        Still needs to be improved. The literature review is insufficient, and does not highlight the innovation of this study.

-        Results

-        Line 93-94:  I suggest that Figs 1A, 1B, and 1C be presented as tables with separate percentages of various pathogens and pests along with some recent references rather than pie graphs.

-        Line 119 What are the possible causes of increased disease incidence with the use of biochar, and can you provide some examples?

-        Line 136 Please include some research on the suppression of nematodes and pests through the use of biochar.

-        Line-147 It would be preferable to draw a correlation matrix rather than simply providing correlation coefficient values (r)

-        Line 149-150 what were applications rates for plant parasitic nematodes, viruses and pests?

-        Line 182-182 It would be better to discuss about how ISR, SAR, and Si work in more details.

-        Discussion:

-         Add few more recent references related to this work.

-        Line 382 As the provided literature does not sufficiently highlight the impact of biochar on phenarogamic parasites, please add some more findings about phenarogamic parasites.

-        Conclusion

-        Conclusion should be climax of all the headings mentioned in the Manuscript. It needs to be modified accordingly.

Reviewer 2 Report

The manuscript draft I have read is well written. I think this article will be an important source to future investigations on use of biochar in agriculture, ecology and biotechnology. 

1. The main goal of this paper is the review for the four aspects that would affect disease suppression and pest control: (i) type and quality of production feedstocks, (ii) temperatures and application rates and (iii) we provide an analysis of mechanisms behind biochar suppressiveness. 2. It is interesting and relevant very much. 3. It is original review for the last 7 years (from 2015). 4. Authors evaluated as literature review how different biochar production feedstock, pyrolysis temperature, application rates, and the pathosystems studied affected disease and pest incidence. 5. The paper is well written. 6. The text is easy to read. 7. The conclusions are consistent with the evidence and the arguments. 8. They do address the main posed question - how different biochar production feedstock, pyrolysis temperature, application rates, and the pathosystems studied affected disease and pest incidence as the literature review.

So it is possible to publish in Plants (MDPI).

Author Response

We thank the reviewer for his positive comments and for appreciating the efforts done over this review. 

Round 2

Reviewer 1 Report

Now it is ok